# Identifying Developmental Language Disorder in Deaf Children with Cochlear Implants: A Case Study of Three Children

**DOI:** 10.3390/jcm12175755

**Published:** 2023-09-04

**Authors:** Gemma Hardman, Rosalind Herman, Fiona Elizabeth Kyle, Susan Ebbels, Gary Morgan

**Affiliations:** 1Department of Language and Communication Science, City, University of London, London EC1V 0HB, UK; gemma.hardman2@nhs.net (G.H.); r.c.herman@city.ac.uk (R.H.); 2Deafness, Cognition and Language Research Centre (DCAL), University College London, London WC1E 6BT, UK; 3Moor House Research and Training Institute, Moor House School & College, Oxted RH8 9AQ, UK; ebbelss@moorhouseschool.co.uk; 4Language and Cognition, Psychology and Language, University College London, London WC1E 6BT, UK; 5Psychology and Education Department, University Oberta Catalunya, 08035 Barcelona, Spain; gmorgan0@uoc.edu

**Keywords:** children, deaf, developmental language disorder, language development, cochlear implant, variability, protective and risk factors

## Abstract

(1) Background: While spoken language learning delays are assumed for deaf and hard of hearing (DHH) children after cochlear implant (CI), many catch up with their hearing peers. Some DHH children with CIs, however, show persistent delays in language, despite protective factors being in place. This suggests a developmental language disorder (DLD). However, at present there is little consensus on how to diagnose DLD in DHH children. (2) Methods: Given the lack of consensus in this area, a set of case studies provides an appropriate first step. The goal of this paper is to show the plausibility of a DLD diagnosis, following careful analysis of protective and risk factors. A retrospective case study review was conducted for three children. Their long-term language outcomes up to four years after CI were considered in the context of access to sound, speech sound discrimination, social skills and non-verbal cognition. (3) Results: It was possible to posit DLD in one child who had experienced good access to sound, alongside good speech discrimination abilities and social development, and normal non-verbal cognition, but who presented with severe language learning difficulties. (4) Conclusions: Finding markers for DLD in DHH children is important for diagnosis and intervention. The implications for clinical practice are discussed.

## 1. Introduction

Deafness occurs in one per 1000 live births [1]. A small number of DHH children are born to DHH parents and acquire a sign language as a native language. The vast majority (90–95%) of children born deaf have hearing parents who do not know any sign language and opt for their DHH child to gain access to sound and spoken language [2]. In the early period, hearing parents struggle to communicate with their DHH child, and consequently language development delay is common. The spoken language delays of DHH children have decreased in the last two decades since earlier identification of deafness has been possible via newborn hearing screening and subsequent early interventions [3,4]. Advances in CI technology have also led to better spoken language development outcomes for DHH children, although there is still variability [5]. Some DHH children catch up with peers quickly, some progress more slowly, and a proportion have persistent and severe delays [6,7,8,9,10,11,12,13,14]. For example in the Geers, Nicholas, Tobey and Davidson [9] sample, a third of children presented with persistent language delay. Our question is, why do some DHH children continue to display such severe delays in language? Might some children have a concurrent developmental language disorder (DLD)? Finding markers for DLD is significant for diagnosis and intervention for DHH children.

## 2. Developmental Language Disorder in Hearing Children

DLD in hearing children leads to serious impacts on everyday interactions and school learning. Reports of the prevalence of DLD in typically hearing children are around 7–10% of the population, covering multiple languages and cultures [15]. DLD can affect children’s abilities in language comprehension (receptive language) and verbal communication (expressive language). Children with DLD also have elevated risks of poorer academic achievements [16].

In the past, research on DLD described hearing children who presented with disordered language skills but had relative strengths in other areas of development or non-verbal skills. Researchers termed this specific language impairment—SLI [17]. For example, many children with SLI were reported to have marked difficulties with sentence repetition [18]. Another typical clinical marker in children exposed to English was difficulty with use and understanding of tense markers, such as “-ed” for past tense [18,19]. SLI was defined by exclusionary criteria, and as such, children with a sensory deficit (i.e., deafness) were excluded from a diagnosis.

More recently there has been a move away from diagnosis by exclusion. Currently, a language disorder is diagnosed purely by inclusion (language difficulties that affect functioning), and then this is split by whether or not there are differentiating conditions. There is growing consensus that for children with DLD, areas other than language might also be affected, e.g., memory. The use of the new term DLD therefore now encompasses a wider group of children with severe language difficulties and delays in related areas of communication and cognition [20].

Where does this broadening of diagnosis leave DHH children with persistent and severe delays in language development? The differentiating conditions inherent in the DLD concept are not meant to be exclusionary criteria, but they can prevent DLD being diagnosed in DHH children. In phase 2 of the CATALISE project [20], the widened category of DLD was used to describe DHH children who experienced the protective factor of early exposure to a sign language but had severe delays in acquiring signs [21]. Thus, DLD was used if the DHH child had early and rich exposure to sign language but had late development and marked difficulties with sign language acquisition. However, for DHH children with persistent delays in spoken language, despite a range of protective factors being present (including normal non-verbal IQ and early aided hearing), a diagnosis of DLD was not considered appropriate. Instead, a diagnosis of language disorder associated with deafness was suggested [20]. If 7–10% of hearing children have DLD, we could probably assume that by chance the same proportion of DHH children might also have DLD. There is no reason to suspect that being deaf would protect you from having DLD. While it was acknowledged in Bishop et al. [20] that a proportion of DHH children may have a heritable, genetic risk factor for an intrinsic language disorder, the current terminology associates their language delays with their deafness (i.e., difficulties with hearing or problems with the CI surgery) rather than considering DLD.

The exclusion of DHH children is despite studies showing these children’s language difficulties have a surface manifestation similar to hearing children with DLD [22]. The Hawker et al. study [22] compared DHH children with expected progression after implant (control group) with six participants who, 7.5 years after CI, continued to present with what the authors termed a disproportionate language impairment in the context of indicators predictive of more typical performance. The control group were matched on aetiology, age at implantation, and CI experience. All children completed a test battery used to identify SLI in normally hearing children. The DHH language impaired children had normal non-verbal IQ and early aided hearing. Scores for the language impaired DHH children were significantly lower than controls on speech perception, articulation, vocabulary, sentence and grammar understanding. Hawker et al. [22] concluded that DHH children can have a language disorder that was separate from both their deafness and their CI, and that this becomes apparent once they have long-term access to useful auditory information and consideration of the presence of indicators predictive of more typical performance. More recently, Sundström, Löfkvist, Lyxell and Samuelsson [23] investigated similarities and differences in phonological and grammatical production between Swedish DHH children, hearing children with DLD, and typical controls, all aged between 4 and 6 years. There were few differences between the two clinical groups, who performed significantly below the controls. However, DHH children found the use of grammatical markers for noun-adjective agreement more challenging than the hearing DLD group.

Thus, a small body of research suggests that DLD may exist in DHH children, but in order to arrive at a diagnosis there are several protective and risk factors which should be considered first. These relate to the intrinsic and extrinsic conditions necessary for language development, and overlap with what Hawker et al. [22] termed indicators predictive of more typical performance.

### 2.1. Protective and Risk Factors for Language Development in Deaf Children

There is much individual variation in language development following CI. Studies have described a range of factors which contribute to this variance: age of implant [4]; cognitive abilities [24,25], parental factors [26,27], and quality of language input [28]. In terms of diagnosing DLD in the current study, we focus on the assessment of four main areas, two of which are extrinsic (present outside the child) in nature: (1) implant age and use of cochlear implant and (2) parent–child interaction. The other two are more intrinsic factors (present within the child): (3) speech sound discrimination and (4) non-verbal cognitive ability. We define each factor and describe how they are measured in clinical settings. Our proposal is that, when faced with cases of persistent and greater than expected spoken language difficulties, analysis of each variable can lead to a diagnosis of DLD rather than language disorder associated with deafness. 

#### 2.1.1. Variations in Age at Implant and Use of Cochlear Implant

Many studies indicate that implantation prior to 12 months of age is advantageous for language development [26,29,30]. McGregor and Goldman [31] argue that early CI coincides with the auditory cortex having the most neural plasticity. Several studies support this argument, for example in a large prospective study, Ching, et al. [24] reported that 5-year-old children who received CIs at 24 months of age had mean language scores 1.4 standard deviations below those of children who received CIs at 6 months of age. In other studies, however, age of implant has been found to have less contribution to outcomes than previously reported, especially among children whose implantation is relatively early [32,33]. Lund [34] for example, showed in a meta-analysis that age of implant was not associated with vocabulary level.

DHH infants also differ in the amount of time their implant is switched on during the day (an extrinsic factor akin to amount of language input). Park et al. [35] reported full-time use of a device was more important than age of implant. Indeed, clinicians recommend that children wear their processors during all waking hours. Less than three hours per day can negatively impact language outcomes [36]. Developments in data-logging technologies have made it easier to identify consistent device users. Apart from age of implant and amount of CI use, there are anatomical and surgical factors that limit the potential benefit of an implant. For example, uncertainty over the cochlear nerve [37] or incomplete insertion of the device [38] can reduce access to sound. Finally the implant function can also change over time if not regularly checked [9]. For this factor, DLD can be considered if a child has persistent and greater than expected difficulties with language, even though there is an early age of implant, high frequency of use and no anatomical reasons why the CI is working at a less-than-optimal level.

#### 2.1.2. Variation in Parent–Child Interaction

The second extrinsic factor to be considered in the context of a DLD diagnosis is the quantity and quality of parent language input to the DHH child. It is the case that input quantity and quality lead to variability in language development in the hearing population [39,40]. Studies by Suttora et al. [41] and Conway et al. [42] showed the role of mother–child interaction and maternal communicative–linguistic input quality in the linguistic development of late talkers. However, hearing children receive language input from many other sources than just parent–child interaction. Indeed, in a recent systematic review, parent input variation was considered a low risk factor for developing DLD in hearing children [43]. Low-quantity and -quality language stimulation is more frequent for DHH children in the first year of life, because deafness precludes children from perceiving much of the parental and surrounding language. It is not until aiding technology is implemented that overhearing surrounding incidental language becomes possible [44,45,46]. The Holzinger’s et al. [46] systematic review and meta-analysis of 27 studies of children with CI reported that the effect of parental linguistic input, including all language types they exposed their children to, accounted for almost one-third (31.7%) of the variance in the children’s language outcomes. The authors concluded that high-quality parent–child language interactions after CI surgery are important for successful language acquisition.

Much research documents how deafness interrupts parent–child interaction [26,27,28]. Both Levine et al. [45] and Dilley et al. [47] made the important observation that parents adapt the complexity of their interactions to the communication skills of the DHH child. Ambrose et al. [48] reported that mothers of implanted children use less complex language, smaller mean length of utterance and fewer different word types. Fagan, Bergeson and Morris [49] found mothers of CI children used more directives and prohibitions. A recent study of all day recordings by Arjmandi, Houston and Dilley [50] showed that DHH children with CIs experienced large individual variability in both the quantity and quality of adult language input.

In turn, the DHH child’s social abilities can influence the level of parental language stimulation offered. DHH infants and children with better social cognitive skills spend more time in language interaction with their parents. In Kelly et al. [51], the early communicative behaviours of DHH infants whose parents were hearing were compared with matched, typically hearing dyads. DHH infants both produced fewer pre-linguistic communicative behaviours during interaction and were less likely to attend to their parent’s reinforcement of the child’s attempt to communicate. A recent study by Hardman, Kyle, Herman and Morgan [14] argued that young DHH children with better pre-linguistic communicative behaviours provoked more stimulating language interactions with their parents.

#### 2.1.3. Variation in Speech Sound Discrimination

The next two factors are more intrinsic to the child. Research shows some hearing children with DLD have speech sound discrimination difficulties e.g., [52]. Deaf children differ in their ability to discriminate between sounds across the speech frequencies [53]. In the clinical context, these speech abilities have different terminologies attached to them e.g., listening skills or auditory comprehension. The current paper focuses on the establishment of speech sounds as the foundation of future language development. Some research refers to speech sound discrimination as ‘hearing function’. For example, Novogrodsky, Meir and Michael [54] showed that DHH toddlers with better hearing function showed syntactic abilities that were similar to age-matched hearing peers. Because speech sound discrimination in noisy environments is challenging [24], DHH children may have fewer cumulative hours of speech processing [55]. If a DHH child is not making the expected progress with language, it is important to evaluate speech sound discrimination and the auditory intervention scheme prior to considering DLD. Clinicians regularly assess listening skills post-implant using tools such as the Nottingham Auditory Milestones (NAMES) checklist [56]. Children who accurately discriminate sounds and progress through the hierarchy of auditory development are expected to have good language development [57].

#### 2.1.4. Non-Verbal Cognitive Ability

Non-verbal cognition is related in several ways to language development in all children. For example, basic attentional control during the first year facilitates word segmentation and the start of intentional communication [58]. The emergence of inhibitory control and working memory in the same early period aids the child to build robust phonological representations and predicts receptive vocabulary at 6 years [59]. Intrinsic variations in non-verbal cognitive abilities can be behind difficulties in language development for some DHH children [8,24,27]. Bishop et al. [20] suggest a diagnosis of DLD can be given to children with lower non-verbal abilities, provided they do not meet the criteria for an intellectual disability (traditionally described as having a non-verbal IQ below <70), but now includes a greater emphasis on personal independence and adaptive reasoning [51]. Low non-verbal IQ is more frequent in DHH than hearing children, especially those children born to hearing parents and whose deafness is not hereditary [25]. Among DHH 3-year-old children in Cupples et al., additional disabilities accounted for 15% to 21% of variance in language outcomes [60]. Nonverbal cognitive ability was significantly correlated with receptive and expressive language outcomes in DHH children who received CIs, and was positively correlated with all language and speech measures [60]. It is therefore important to have an assessment of non-verbal cognitive skills in order to better understand the association of DHH children’s persistent language delay.

In summary, using these four indicators of typical performance we can understand why there are different profiles of language development in DHH children. Those who (1) will catch-up with their hearing peers; (2) present delays in language linked to poor functioning in one or more areas of implant use, speech sound discrimination, poor parental interaction or non-verbal cognition; and (3) present with persistent language delays in the presence of relatively good development in the four factors. Consideration of these factors will enable us to posit DLD. The rest of the paper discusses three case studies which illustrate these different profiles in more depth.

## 3. Materials and Methods

A retrospective case note review was conducted for three DHH children who received bilateral CI between 14 and 24 months of age. All were known to a UK tertiary level CI programme, and were selected for the purpose of comparison within this report. The children were purposefully selected from a large clinical sample (80–100 cases per annum) in order to illustrate different profiles of language development commonly seen in CI centres. The three children were comparable in gender, age, deafness aetiology, pre-CI audiological levels and family hearing status. They differed in the impact of variability of the four indicators of typical performance reviewed in the current paper. Parental consent was obtained for participants’ histories to be audited and reported. Given this is a retrospective analysis and that implant technology has changed substantially over the last decade, it is important to point out that the children received CIs in the last 10 years. The first author is a clinician in the CI programme the children were attending. There were a small number of missing data points across the 5-year records, due to missed appointments.

### 3.1. Participants

It is important to acknowledge that these participants are children in a real clinical service rather than in a research study. Therefore, there are complexities in their backgrounds and assessment that make clear categorisation of their profiles more difficult. However, a detailed description of their development against the four main indicators predictive of typical performance is useful. All three children were born to hearing parents in the last 10 years, and deafness was identified at newborn hearing screening. All were exposed to spoken language, supplemented with some lexical signs used alongside spoken language. No formal sign language assessment was carried out during the pre- and post-implant follow-up. Key demographic information for the participants is in Table 1. All children had a full insertion of their internal implant. None had a significant visual impairment, or any known neurological dysfunction or oral–motor difficulties sufficient to affect spoken language. All children were seen for monthly speech and language therapy (SLT) appointments at the tertiary centre for the first year post implant, in addition to receiving local support from SLT and peripatetic teaching services. 

### 3.2. Implant Use

Details of pre-implant audiological testing can be found in Table 1. All three children were born with a bilateral severe–profound hearing loss and met assessment criteria for bilateral CI. Children attended regular audiology appointments following CI, where the audiologist programmed their devices based on the individual child’s auditory responses to electrical stimuli in a process known as mapping. Hearing levels were reviewed each year post CI, depending on the child’s compliance and attendance. Aided thresholds were obtained using soundfield audiometry. Soundfield audiometry assesses a child’s hearing responses when acoustic signals are presented through one or more sound sources in the room.

### 3.3. Speech Sound Discrimination

These skills were assessed using two parent/carer reported-outcome measures: the Categories of Auditory Performance II (CAP-II) [61] and the NAMES [56]. The CAP-II asks parents/carers to rate their child’s speech sound discrimination when wearing their hearing aids or CI on a scale from 0 (no awareness of environmental sounds) to 9 (use of telephone with an unknown speaker in an unpredictable context). The NAMES is a tracking tool designed to monitor listening progress following CI, and was recently validated for use in children implanted under two years of age [56]. The Test of Auditory Perception of Speech for Children (TAPS-II) [62] and a Speech Intelligibility Rating (SIR) [63].

### 3.4. Non-Verbal Cognition and Parent–Child Interaction

Prior to implantation, all children were screened using the Schedule of Growing Skills II (SGS-II) [64] by a clinical psychologist. The SGS-II screens for strengths and weaknesses across a variety of domains, providing an approximate developmental age (based on hearing children) for each. The Leiter Revised International Performance Scale (Leiter-R) [65] was also used because it requires no verbal instructions for administration and response, making it suitable for DHH children. The test has two scales, visualisation and reasoning, containing subtests that assess a variety of abilities including spatial perception, problem solving, and attention to detail. Scaled scores of 8–12 and standard scores of 90–109 are in the average range.

### 3.5. Language Testing

Formal language testing was completed at yearly intervals post-implantation using measures appropriate for the child’s age and ability. The Preschool Language Scales, fourth and fifth editions (PLS-4, 5) [66,67] is used from birth to 7 years 11 months, to assess a range of language skills, and each test has guidelines for use with DHH children. Standard scores were provided for the auditory comprehension and expressive communication subscales and for total language (a combination of both scales). 

The Preschool Clinical Evaluation of Language Fundamentals, Edition 2 (Preschool CELF-2) [68] is designed for children aged 3–6 years. Subtests have a mean scaled score of 10 and a standard deviation of 3, i.e., scores of 7–13 are within the normal range. The Clinical Evaluation of Language Fundamentals, Edition 4 (CELF-4) [69] is a standardized language assessment with norms based on English monolingual hearing children from 5 to 16 years 11 months. It includes a sentence-repetition task, which is a good clinical marker of DLD. As with the Preschool CELF-2, CELF-4 scaled scores between 7 and 13, are within the normal range.

## 4. Results

### 4.1. Case Study A

Child A’s bilateral profound hearing loss was identified following newborn hearing screening with an unknown aetiology.

#### 4.1.1. CI Use and Access to Sound

Bilateral CIs were activated at 18 months, worn for all waking hours, and regularly maintained (see appointment attendance, Table 2). By two years post implant, her soundfield aided thresholds were approximately 30 dBA across frequencies 250 Hz–6 kHz. By six years post implant, aided performance testing indicated she had access at 20–25 dBA across frequencies 500 Hz–4 kHz (see Table 2). This indicates that there were no concerns relating to the variable CI use and access to sound.

#### 4.1.2. Speech Discrimination Skills

Child A met all expected milestones by one year post implant, as measured by the NAMES [56]. At two years post implant, she was observed to be listening naturally to spoken language and could follow spoken instructions without needing to look at the speaker. By three years post implant, she could repeat most words within sentences (e.g., 61/64 words were repeated correctly on the TAPS-II) [62]. She used the telephone with a known speaker using listening alone (i.e., without access to lip patterns, visual cues or gesture). There were no concerns relating to speech discrimination or speech intelligibility, as illustrated by the increasing CAP-II and SIR ratings across the years (Table 2).

#### 4.1.3. Non-Verbal Cognitive and Social Skills

Prior to CI, Child A’s developmental milestones were noted to be appropriate for her age as measured by the SGS-II and paediatrician report. Her motor milestones were developing typically, and there were no concerns relating to her non-verbal abilities. During her CI assessment period, Child A was observed to have age-appropriate pre-linguistic communication and social interaction skills, including those not reliant on hearing, such as use of gesture, eye gaze, joint attention and pretend play. She presented as an alert, communicative and sociable baby. Following CI switch-on, Child A received regular visits (weekly to twice weekly) from her local team of professionals in both education and health. She attended a local English-speaking mainstream nursery. In her second year post implant, Child A returned to her home country, but continued to be exposed to English and Hindi. By three years post implant, she was attending a mainstream school with a hearing impairment unit (using an auditory–oral approach) and receiving regular support from a teacher of the deaf (TOD) and an SLT. She attended this school throughout her primary years, spending increasing amounts of time within the mainstream classroom as her speech discrimination and language skills progressed. She developed positive friendships with both hearing and DHH peers.

#### 4.1.4. Language Development

Child A was exposed to spoken English and Hindi at home, although her family chose to communicate with her primarily in English. There are several issues with assessing a bilingual child’s language, and this is compounded by the added complexity of the deafness. It is possible for a child with CIs to develop proficiency in both a first and second language, although they may develop their second language at a slower rate than their typically hearing peers e.g., [70]. Although Child A made progress with language, her language skills remained delayed compared to her monolingual hearing peers for the first two years post implant (see Table 2). However, by three years post implant, she had closed the gap between her language levels and those of her hearing peers, with most scores on formal language assessments (in English) falling within the average range for her age. She continued to have a mild delay with vocabulary development. Her speech was rated as intelligible to a listener with limited experience of a DHH person’s speech. Child A’s progress was therefore judged to be typical by the CI team for a child with her history, as she was catching up with her hearing peers.

### 4.2. Case Study B

Child B’s bilateral severe–profound hearing loss was identified following newborn hearing screening, secondary to Connexin 26 mutation. This is a non-syndromic genetic cause of hearing loss that usually does not have associated difficulties. Bilateral CIs were activated by 24 months of age, and the home language was English.

#### 4.2.1. CI Use and Access to Sound

This child only attended 45% of appointments in the first year, and therefore her devices were not maintained regularly. Aided performance testing indicated she had access to sounds across the speech frequencies at 30–40 dBHL (tested annually post implant), although she had difficulties establishing consistent processor use for the first four years post implant. At approximately six years of age, child B’s device use improved.

#### 4.2.2. Speech Sound Discrimination

Child B’s ability to understand the sound signal and follow spoken language without the aid of lip-reading was limited. By three years post implant, her device use had improved but continued to be inconsistent. When wearing the CIs, Child B could detect all tested speech sounds and discriminate between four to six speech sounds. She was able to discriminate between one- and three-syllable words, but found it difficult to discriminate between one- and two-syllable words, or two- and three-syllable words. For many words, she was reliant on lip patterns and lexical signs, with spoken English. Following more consistent device use, Child B’s speech discrimination skills improved, but speech intelligibility (SIR scores) remained very poor across the years (Table 3). By five–six years post implant, she was able to identify single words and short sentences, but continued to have difficulty understanding novel instructions.

#### 4.2.3. Parent–Child Interaction

During the pre-implant assessment, Child B demonstrated early positive social interaction, including joint attention and smiling towards family members and professionals. Following device switch-on, Child B’s attendance at both tertiary hospital appointments and with local services was inconsistent. In the early post-implant years, when observed in her mainstream nursery, child B often played alone, and rarely initiated communication with peers or adults.

#### 4.2.4. Non-Verbal Cognition

Child B’s cognitive skills were assessed at varying points prior to and following cochlear implantation (see Table 3). The SGS-II indicated that general development was age-appropriate in all areas, with a particular strength in visual understanding and interactive play skills. Fluid reasoning (the ability to complete patterns and sequences in non-verbal stimuli) was an area of difficulty, falling within the low–average range. However, her full- scale IQ remained within the average range for her age.

#### 4.2.5. Language Development

She attended a mainstream school with a DHH unit with a Total Communication approach (Total Communication includes different modes of communication i.e., signs, spoken language, written or visual aids, depending on the needs of the child), and spent much of her time within small classes in the unit. The family were encouraged by professionals to introduce some signs while speaking, although this was not formally taught or used consistently until primary school entry. By three years post implant, Child B increased in confidence and social interaction, and her communication skills improved with family and friends. By six years post implant, Child B had become a confident communicator with familiar and unfamiliar people, often using a combination of lexical signs, gesture, and spoken-word approximations.

Child B’s progress with spoken language during the first three years following activation of her CI device was limited (see Table 3). Her language scores at two and three years’ post implant placed her on the first percentile compared with her hearing peers. By three years post implant, she used signed sequences alongside word approximations to communicate. Although she made some progress with speech discrimination and spoken language skills in her later years, she continued with sign as her preferred mode of communication. Signing was not formally assessed, but it is noted that she made significantly more progress with this modality than oral language. It is unlikely that DLD selectively impairs spoken but not signed language [51]. It is still possible that DLD is present, but this is clouded by wide variability in all factors outlined previously: inconsistent device use, missed hospital attendance at therapy, disturbances to speech discrimination development, and delays in the establishment of parent–child interaction patterns and early social abilities.

### 4.3. Case Study C

Child C’s profound hearing loss was identified following newborn hearing screening, secondary to an autosomal recessively inherited disorder. 

#### 4.3.1. CI Use and Access to Sound

Bilateral CIs were activated at 14 months of age, and she consistently wore her devices for all waking hours. Aided testing at two years post implant indicated she was responding reliably at 25–30 dBHL across frequencies 500 Hz–4 kHz. By four years post implant, Child C had reliable access to sounds across the speech frequencies at 15–20 dBA (500 Hz–6 kHz). The family attended 90% of appointments within the first five years post implant (see Table 4), and the audiology team judged Child C’s processors were working at an optimal level, with consistent device maintenance.

#### 4.3.2. Speech Sound Discrimination

By two years post implant, Child C was detecting sounds across the speech frequencies and was able to correctly identify single words differing in syllable number and stress pattern. By three years post implant, she could follow familiar instructions (e.g., ‘touch your nose’). However, her ability to complete the sentence-repetition task was poor. While she was able to repeat 54/64 key words in sentences, she frequently missed grammatical markers and inflections. Her CAP-II score was 5 (understanding of common phrases). There were no concerns relating to speech discrimination or speech intelligibility, as illustrated by the increasing CAP-II and SIR ratings across the years (Table 4.) Anecdotally, her mother reported she could hear what was said to her but could not follow conversations.

#### 4.3.3. Social Skills

At her pre-implant assessment, Child C was noted to be an interactive and responsive baby, able to use eye contact and facial expressions (e.g., social smiling) to interact with familiar adults, and was very vocal. She attended a local playgroup until she was three, then attended a nursery attached to a school with a DHH resource base (with an auditory–oral approach). She continued to attend the same school at school-entry age. Child C went on to develop friendships, many with peers who also had language difficulties where their interactions and play often involved games that required minimal language.

#### 4.3.4. Non-Verbal Cognitive Skills

Pre-implant assessment revealed that Child C’s non-verbal skills were age-appropriate. At this time, based on parental report, she had a mild delay in gross motor skills. However, when her non-verbal skills were formally assessed four years post implant at age five years, these were within the average range.

#### 4.3.5. Language Development

Following activation of her CI devices, Child C was noted to be quiet and shy during appointments with the CI team. However, information from local services indicated she was more confident with familiar others, and played appropriately with peers within the nursery, and subsequently school, setting. She received weekly input from either an SLT or TOD, often alternating each week.

As with Child A, Child C was exposed to two languages at home (English and Arabic), although her family communicated with her primarily in English. As with Child A, there are several complicating factors in differentiating language delay and language disorder with a child who is exposed to two languages. Establishing language disorder in a bilingual child requires thorough assessment of both languages; in this case, assessment was only carried out in English, as this was the language the parents reported they communicated in with their child. It is possible, however that she was not able to access the surrounding language if this was in Arabic.

While Child C made progress with her speech sound discrimination, her progress with language was much slower (see Table 4). Concerns were noted at two years post implant by local professionals, regarding her language and communication difficulties, with English language scores at the second percentile. Crucially, Child C’s older hearing sibling was diagnosed with DLD at 5 years of age and her DHH sibling also presented with additional language learning difficulties. As a result, Child C’s mother had received extensive training and support from professionals to improve parent–child interaction.

By three years post implant, her language scores continued on the first percentile and the gap between her language levels and those of her hearing peers became much wider (see Table 4). Her understanding of language was inconsistent, and she was unable to retain words and concepts that had been explicitly taught. She had particular difficulty engaging in conversations when unfamiliar or more abstract vocabulary was used by others. In the light of excellent device function and use, good non-verbal and social abilities, and siblings with diagnosed language impairment, we conclude this child has a DLD in addition to her deafness.

## 5. Discussion

Considerable variation in spoken language outcomes is common in DHH children who receive a CI [7,8,9,10,11,12,13]. Persistent language difficulties are evident among some DHH children, but research and practice has not yet been able to determine if this is DLD. The relevant take-away message is that a diagnosis of DLD in DHH children with persistent language development delay is not appropriate until indicators predictive of more typical performance have first been addressed. Although it is complex to distinguish a co-existing DLD from temporary language delay, in a growing number of studies e.g., [22,23,54] the profiles of DHH children with suspected DLD are very similar to those of hearing children with the same diagnosis. In the current study, such a candidate for a DLD diagnosis is child C, who had good speech sound discrimination and production skills, and social and cognitive abilities. At the same time, her language was very delayed and displayed error patterns similar to those reported in hearing children with DLD. She used short phrases to communicate, tended to use incorrect word order, omit function words such as determiners (e.g., the, a) and auxiliary verbs (e.g., is, was), and had a small verb repertoire (e.g., said ‘fish man’ for ‘the man is catching the fish’). On a sentence-repetition task, three years after receiving an early CI, child C still omitted grammatical markers and inflections. Grammatical errors in sentence repetition is one of the best clinical markers of DLD in hearing children [71,72,73].

The other two DHH children described in the current study also presented wide variations in the indicators of more typical language development [40,54]. Child A represents a late-talking DHH child who subsequently catches up. She received an early CI, which gave prolonged and quality access to spoken language [24,30]. Speech discrimination and production skills were consistently good, indicating both intact cognitive processing of sounds [57] and good CI function [34]. This gave the child a strong foundation for later language development [54,74]. The child displayed appropriate pre-linguistic communication behaviours, including joint attention and spontaneous gestures [14]. This indicates good social abilities and the motivation to take advantage of good parent–child interaction [43,46]. While she had some vocabulary delays in the first 24 months post implant, possibly due to her deafness and exposure to two spoken languages, the following 12 months she caught up with her typically hearing peers, suggesting intact language development.

Child B, on the other hand, displayed a number of delays and disturbances with the same indicators offered. These complications did not resolve, and while this does not necessarily preclude the presence of DLD, several issues would cloud any definitive diagnosis. There were no issues with the aetiology of the deafness [25] but following the CI there were concerns about how much her environment enabled her to take advantage of the intervention. Missed appointments might mean the CI team at the hospital were less able to fine-tune the CI to maximise access to speech sounds [9]. The foundations for language development were thus not in place. Low-quantity and -quality input resulted in continued speech sound discrimination and production difficulties four years post implant. Coupled with this, the environmental support needed to establish consistent parent–child interaction patterns were judged to be inconsistent. Although the child favoured visual forms of communication including signs and gestures, her parents did not learn or use a formal sign language with her, and thus missed opportunities for language learning [75]. Early social interaction was also a concern; it is not clear if this was because the child was not understanding language or because she was avoiding engaging with others [14]. Her early social difficulties resolved in the coming years, yet at 5 years post implant the child was still not understanding instructions. This child presents with a language disorder associated with deafness because of concerns across several indicators linked to hearing and her CI. Doubts across extrinsic and intrinsic factors clouded any diagnosis of DLD as an explanation of her persistent language delays.

There are several clinical implications of this research. The applicability of the new diagnostic label DLD merits consideration [20]. As described in the introduction, currently DHH children learning spoken language are excluded from the diagnosis, despite several other groups of language-delayed children entering the new category of DLD because they have language difficulties that affect wider functioning. If a DHH child is exposed to a signed language early in development and presents with a disorder they can be included as signing DLD [76]. However, DHH children struggling with oral language despite protective factors being in place receive a diagnosis of language disorder associated with deafness. We argue that by using specific diagnostic criteria for DLD in DHH children using CIs we can say with more certainty that deafness and DLD can co-exist.

Moreover, there is clinical value in making a diagnosis of DLD in a deaf child. There is no research on the characteristics of a language disorder associated with deafness, and no specific evidence-based interventions for professionals to turn to. In contrast, a diagnosis of DLD carries greater shared meaning and has more established clinical implications in terms of the amount and type of support likely to be needed (albeit based on a diagnosis of DLD in hearing children), and which may differ from the intervention provided for children without DLD. This differential diagnosis would allow clinicians, parents and local support services to more confidently explore differentiated and more intensively delivered interventions than those traditionally used for DHH children. For example, a systematic review by Frizelle et al. [77] noted the benefit of intervention features for hearing children with DLD, such as the use of explicit rather than implicit instruction and the use of elicited production. These approaches would not be advised for a child with a language delay, for whom better environmental language learning opportunities are more likely to be recommended [78]. 

A diagnosis is also important in terms of prognosis. There are key areas known to be at risk following a diagnosis of DLD. For example, hearing children with DLD are known to encounter greater difficulties with reading, spelling and maths [79] and are more likely to experience anxiety and depression [18] than those without DLD. Where DLD is present in addition to deafness, the potential for these areas to be impacted may be even greater. Finally, a diagnosis of DLD is important in helping parents to better understand the nature of their child’s difficulties [80] and why they may struggle more to learn language than other DHH children. This can also be helpful for parents/professionals in understanding that the child’s trajectory for language development is likely to be different to that of a child with a language disorder associated with deafness, and to thus adjust their expectations for support and intervention; these children are likely to require higher levels of support from all services in education and at home.

Before formalising the diagnostic value of the approach proposed, it may be helpful to conduct a Delphi consensus study to reach agreement on terminology and criteria. This could include both theoretical and clinical implications of including DHH children in the diagnosis. In CI centres, professionals are often asked what can be done about a child with a CI who is receiving good support but does not develop language as expected. By re-framing a child’s difficulty in terms of a neurocognitive disorder of language learning rather than a perceptual deficit, professional intervention is recalibrated.

Although the proposed set of variables was useful for considering DLD in the cases considered, they are retrospective rather than predictive. We suggest that the method we have set out is important in initiating conversations with professionals about the support best suited to DHH children with DLD. Although it is important to be prudent with findings based on small numbers, the current study has implications for teachers and SLTs. These case studies demonstrate that by four years post implant it is possible to identify different language trajectories. However, detailed information about a child’s background and early experiences is required, in order to differentiate between DHH children who have language delay as a result of disturbances to extrinsic/intrinsic variables and those who have DLD. Lastly, the protective/risk factors explored in the current study are based on information that is routinely collected by clinicians. It would be helpful for clinicians working with DHH children to be aware of their potential diagnostic importance, to ensure that data are collected and considered when evaluating DHH children’s progress and for decisions about education placement and the support needed to maximise learning. 

There are some limitations with the study. As described previously, the three children were typical in their heterogeneity. CI Centres in the UK report that 28% of children receiving CIs are from families where the home language is a spoken language other than English [81]. Two of the three children were exposed to two spoken language in different degrees and at different points in development. One child had typical English development within 12 months of implant (Child A) while the second had persistent language delays five years post implant (Child B). A recent review of 22 studies reported that, whilst there was a high degree of variability, there was no adverse effect of bilingualism for DHH children [82]. While DHH children can learn two spoken languages, there are factors at play for each individual child that can impact on the quantity and quality of exposure to each language [83]. Children who are DHH with persistent language delays, and have variable exposure to different spoken languages during the pre- and post-implant period can make DLD diagnosis more difficult. The retrospective nature of this case report has weaknesses because some data were not available in sufficient detail. For example, whilst all three families identified English as the child’s primary language, input was not quantified. Composite scores for language were not available within case notes, and so it was only possible to report language assessment subtest scores. Finally, the three cases only included DHH children with CIs, rather than hearing aid users, which should be considered in future studies [84,85].

## 6. Conclusions

This paper reviewed three cases of language delay in DHH children with CIs, and after careful consideration of a range of factors narrowed down one case of DLD. Larger scale, longitudinal studies are also important to investigate the plausibility of using this method. This is necessary, given the complexity of their presentation, and may also provide further indications of prevalence within the DHH paediatric population. A diagnosis of DLD has potential benefits for individuals in terms of the intervention they receive and parental understanding of their child’s condition.

## Figures and Tables

**Table 1 jcm-12-05755-t001:** Key demographic information.

	Child A	Child B	Child C
Gestational age and birth history	42 weeks, emergency C-section due to lack of progress with labour and foetal distress.	39 weeks, normal pregnancy. Mild jaundice for one week.	40 weeks, normal pregnancy, no concerns after birth.
Gender	Female	Female	Female
Cause of deafness	Unknown (congenital)	Connexin 26	Autosomal recessive disorder
Pre-implant audiology levels (unaided)	95 -> 115 dBHL	95 -> 120 dBHL	95 -> 110 dBHL
Radiological findings	Normal anatomy of inner ear structures. Cochlear nerves present bilaterally.	Normal anatomy of inner ear structures. Cochlear nerves present bilaterally.	Normal anatomy of inner ear structures. Cochlear nerves present bilaterally.
Languages spoken at home	English and Hindi	English	English and Arabic
Primary language used with child	English	English	English
Frequency of local speech and language therapy input in first year post-implant	Twice weekly	Twice weekly	Twice weekly

**Table 2 jcm-12-05755-t002:** Child A’s test results.

	Pre-Implant	1-Year Review	2-Year Review	3-Year Review	5-Year Review
**Age of Testing**	11 months	2 years	3 years	4 years	6 years
**CAP-II Score**	0	5	7	7	9
**SIR Score**	1	2	3	4	5
**Language Test**	Not used	PLS-4	PLS-4	Preschool CELF 2	CELF-4
**Language Results**	Understanding some single signs	Auditory Comprehension: Standard score: 79 Percentile: 8 Expressive Communication: Standard score: 85 Percentile: 16	Auditory Comprehension: Standard score: 64 Percentile: 2 Expressive Communication: Standard score: 65 Percentile: 1	Sentence Structure: Scaled score: 10 Word structure: Scaled score: 9 Expressive vocabulary: Scaled score: 5	Concepts and following directions: Scaled score: 11 Word structure: Scaled score: 12 Expressive vocabulary: Scaled score: 8
**Non-Verbal Assessment**	SGS-II at 11 months (Skill: age equivalent in months)				
**NV Results**	Active postural: 12Locomotor: 15 Manipulative: 15 Visual: 10 Interactive social: 15 Self-care social: 15 Cognitive: 10				
**% appointments attended**	100% (6/6)	100% (16/16)	100% (4/4)	100% (2/2)	100% (2/2)
**Aided Testing** **500 Hz–4 kHz**	Not tested	30 dBA	Not tested	30–35 dBA	15–25 dBA

**Table 3 jcm-12-05755-t003:** Child B’s test results.

	Pre-Implant	1-Year Review	2-Year Review	3-Year Review	5-Year Review
**Age of Testing**	17 months	3 years	4 years	5 years	7 years
**CAP-II Score**	0	0	2	4	5
**SIR Score**	1	1	1	1	1
**Language Test**	Not tested.	Not tested.	PLS-4	PLS-4	Not tested.
**Language Results**	Understanding small number of signs and gestures.	Not wearing processors consistently. Understanding simple sign strings. Copying lip patterns (without voice) and signs, using single signs or gesture to communicate.	Auditory Comprehension: Standard score: 55 Percentile: 1 Expressive Communication: Standard score: 55 Percentile: 1	Auditory Comprehension: Standard score: 55 Percentile: 1 Expressive Communication: Standard score: 55 Percentile: 1	BSL dominant. Limited vocabulary and delayed BSL, based on informal assessment.
**Non-Verbal Assessment**	SGS-II at 17 months (Skill: age equivalent in months)	Leiter-R	Leiter-R		
**NV Results**	Locomotor: 15 Manipulative: 18 Visual: 24 Interactive social: 24 Self-care social: 18 Cognitive: 18	**Full-scale IQ:** Standard score-129 Percentile: 97 Category: high	**Full-scale IQ:** Standard score: 106 Percentile: 66 Category: average		
**% appointments attended**	55% (6/11)	45% (10/22)	42% (5/12)	63% (5/8)	100% (5/5)
**Aided Testing** **500 Hz–4 kHz**	Not tested	Not tested	35–40 dBA	30–40 dBA	25–35 dBA

**Table 4 jcm-12-05755-t004:** Child C’s test results.

	Pre-Implant	1-Year Review	2-Year Review	3-Year Review	4-Year Review
**Age of Testing**	7 months	2 years	3 years	4 years	5 years
**CAP-II Score**	0	4	4	5	6
**SIR Score**	1	2	3	3	3
**Language Test**	Not tested.	PLS-4	PLS-5	PLS-5	Preschool CELF 2
**Language Results**	Responsive to her routine, demonstrating likes and dislikes. Using long vowels.	Auditory Comprehension: Standard score: 66 Percentile: 1 Expressive Communication: Standard score: 83 Percentile: 13	Auditory Comprehension: Standard score: 67 Percentile: 1 Expressive Communication: Standard score: 75 Percentile: 5	Auditory Comprehension: Standard score: 70 Percentile: 2 Expressive Communication: Standard score: 66 Percentile: 1	Sentence Structure: Scaled score: 3 Word structure: Scaled score: 1 Expressive vocabulary: Scaled score: 2
**Non-Verbal Assessment**	SGS-II at 6 months (Skill: age equivalent in months)				Leiter-R
**NV Results**	Passive postural: 3 Active postural: 1 Manipulative: 6 Visual: 6 Interactive social: 6Self-care social: 6				**Full-scale IQ:** Standard score: 99 Percentile: 47 Category: average
**% appointments attended**	100% (6/6)	94% (16/17)	83% (5/6)	100% (4/4)	100% (2/2)
**Aided Testing** **500 Hz–4 kHz**	Not tested	30–35 dBA	25–30 dBA	20–30 dBA	15–25 dBA

## Data Availability

Original data are unavailable due to privacy restrictions.

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
