# Peer review of "Identifying Developmental Language Disorder in Deaf Children with Cochlear Implants: A Case Study of Three Children"

_jcm, 2023, doi:10.3390/jcm12175755_

Round 1

Reviewer 1 Report

Thank you very much for giving me the opportunity to review this very interesting manuscript. With some improvements the manuscript has the potential to be informative and comprehensive in the field of Developmental Language Disorder in Deaf Children with Cochlear Implants.

At overall the manuscript is generally well-written, organized and appropriate for the scope of the Section Clinical Pediatrics. The style of writing is appropriate for a scientific review, although some sentences could be rephrased for improved flow and readability.

Regarding the overview of the study, I would like to highlight some issues that have been identified and related to the preparation of the manuscript https://www.mdpi.com/journal/jcm/instructions:

·       regarding word limits the manuscript as a whole as well as the abstract word limit appear to have been exceeded. Please revise the manuscript with the appropriate instructions for authors (200 words).

·       The abstract should be shortened and brief and add the (1), (2), (3), (4) numbers in the abstract categorization.

·       Please in the key words add more keywords to improve your manuscript visibility.

Introduction

The introductory section of this manuscript presents a comprehensive overview of the subject matter pertaining to Developmental Language Disorder (DLD) in children with normal hearing abilities. This study underscores the significant consequences of Developmental Language Disorder (DLD) on routine social exchanges and educational attainment, underscoring its detrimental effects on children's language comprehension, verbal expression, and academic achievements in children. In lines 34 -42 you made a good explanation about the criticalness, and you set well aimed question. Overall, the introduction provides a clear and concise overview of the topic, highlighting the shift from SLI to DLD and the potential existence of DLD in DHH children. It effectively sets the stage for the subsequent discussion.

2. Developmental Language Disorder in Hearing Children

The review of protective and risk factors for language development in DHH children is well-presented and informative. However, there are a few suggestions and issues to consider.

In some cases, the inclusion of supplementary citations may be necessary to substantiate the assertions and arguments put forth in the manuscript. This approach would enhance the validity of the arguments and contribute to a more thorough examination of the current body of the existing literature. Regarding the Clarity of language, the majority of the manuscript is clear and concise. Although, there are a few instances where the language and sentence phrasing could be further clarified or simplified to enhance understanding and to ensure smoother reading.

3. Methodology

Please elaborate on the specific criteria used for the selection process. The section mentions that three children were chosen for comparative analysis. However, it would be beneficial to provide further clarification regarding the specific criteria employed in the selection process of these participants. Possible variables such as age, gender, language background, or other characteristics.

4. Results

The section results are well-written and well-presented. Very interesting information is presented in detail.

5. Discussion

The Discussion should begin by summarizing the main findings of the study and relating them to the research questions or objectives stated earlier. This will provide a clear overview and help the readers understand the importance of the findings.

Please, it would be valuable to acknowledge and address some potential alternative explanations for the language difficulties observed in the DHH children by factors such as cognitive abilities, socio-environmental influences, or other comorbidities that may contribute to language outcomes.

The study primarily focuses on children with cochlear implants.  However, it is advisable to recognize the significance of incorporating children with hearing aids in forthcoming research studies by recognizing the potential variations and commonalities in language outcomes among these two groups.

6. Conclusion

Please reform the section and conclude with a concise summary of the main points.

Author Response

Thank you very much for giving me the opportunity to review this very interesting manuscript. With some improvements the manuscript has the potential to be informative and comprehensive in the field of Developmental Language Disorder in Deaf Children with Cochlear Implants.

At overall the manuscript is generally well-written, organized and appropriate for the scope of the Section Clinical Pediatrics. The style of writing is appropriate for a scientific review, although some sentences could be rephrased for improved flow and readability.

>manuscript reviewed with the intention of improving flow and readability

Regarding the overview of the study, I would like to highlight some issues that have been identified and related to the preparation of the manuscript https://www.mdpi.com/journal/jcm/instructions:

  • regarding word limits the manuscript as a whole as well as the abstract word limit appear to have been exceeded. Please revise the manuscript with the appropriate instructions for authors (200 words)

>We reduced the word count of the abstract to 196 words. We were advised by the editoral support that there is no maximum word count indicated

  • Add the (1), (2), (3), (4) numbers in the abstract categorization.

>Numbered sections added in abstract

  • Please in the key words add more keywords to improve your manuscript visibility.

>We added several more words

Introduction

The introductory section of this manuscript presents a comprehensive overview of the subject matter pertaining to Developmental Language Disorder (DLD) in children with normal hearing abilities. This study underscores the significant consequences of Developmental Language Disorder (DLD) on routine social exchanges and educational attainment, underscoring its detrimental effects on children's language comprehension, verbal expression, and academic achievements in children. In lines 34 -42 you made a good explanation about the criticalness, and you set well aimed question. Overall, the introduction provides a clear and concise overview of the topic, highlighting the shift from SLI to DLD and the potential existence of DLD in DHH children. It effectively sets the stage for the subsequent discussion.

  1. Developmental Language Disorder in Hearing Children

The review of protective and risk factors for language development in DHH children is well-presented and informative. However, there are a few suggestions and issues to consider.

In some cases, the inclusion of supplementary citations may be necessary to substantiate the assertions and arguments put forth in the manuscript. This approach would enhance the validity of the arguments and contribute to a more thorough examination of the current body of the existing literature.

>We have added some extra citations. See bolded references.

Regarding the Clarity of language, the majority of the manuscript is clear and concise. Although, there are a few instances where the language and sentence phrasing could be further clarified or simplified to enhance understanding and to ensure smoother reading.

>We revised the manuscript for clarity and have made several changes to wording to improve the reading.

  1. Methodology

Please elaborate on the specific criteria used for the selection process. The section mentions that three children were chosen for comparative analysis. However, it would be beneficial to provide further clarification regarding the specific criteria employed in the selection process of these participants. Possible variables such as age, gender, language background, or other characteristics.

 > We chose three children as stereotypical examples of the types of clinical and language profiles that are common in the CI setting. We have added some more information explaining this in the text. Bolded pages 5-6 

  1. Results

The section results are well-written and well-presented. Very interesting information is presented in detail.

 > many thanks for this positive feedback

  1. Discussion

The Discussion should begin by summarizing the main findings of the study and relating them to the research questions or objectives stated earlier. This will provide a clear overview and help the readers understand the importance of the findings.

>Done

Please, it would be valuable to acknowledge and address some potential alternative explanations for the language difficulties observed in the DHH children by factors such as cognitive abilities, socio-environmental influences, or other comorbidities that may contribute to language outcomes.

>We have provided more explanations of these variables in the expanded discussion of child B. Pages 11-12 and 14-15

The study primarily focuses on children with cochlear implants.  However, it is advisable to recognize the significance of incorporating children with hearing aids in forthcoming research studies by recognizing the potential variations and commonalities in language outcomes among these two groups.

> this point is covered in the last lines of the discussion p16

  1. Conclusion

Please reform the section and conclude with a concise summary of the main points.

>Done

Reviewer 2 Report

Manuscript ID: JCM-2517724

Identifying Developmental Language Disorder in Deaf Children with Cochlear Implants: A Case Study of Three Children

To some extent, spoken language delays can be assumed for deaf children who receive cochlear implants, but research increasingly indicates that most do catch up with their hearing peers. That said, it is also an important and ongoing concern as to why some do not. The authors question whether these delays, at least in some cases, can be attributed to a developmental language disorder (DLD) that exists along with the hearing loss. They suggest that a diagnosis in this case can be made by considering four protective and risk factors that are both extrinsic (age at implantation and consistent device use, and quantity and quality of the parent-child interaction) and intrinsic (speech-sound discrimination, and non-verbal cognition) (pp. 4-5). As I understand their argument, if the language delay cannot be accounted for by an analysis of one of these factors, then it would be appropriate to posit DLD as an explanation, rather than a language disorder associated with deafness (p.3).  

While this is an interesting proposition, I have questions about the retrospective analysis, and the extent to which a diagnosis of DLD in children with CIs has utility for clinical and pedagogical practice.

Retrospective Analysis

I would like to see the authors say more about how these three case histories were chosen for analysis. What was the total participant pool from which these cases were drawn? What was the process by which these three were identified? Was there any relationship of the authors to this CI programme? As well, it would be important to explain why there is missing data for some time points in some areas (e.g., non-verbal assessments for Child A, Language for Child B in Year 5).

The description of Child A is clear and provides an example of how a child with a CI catches up to hearing peers. The depiction of Child C illustrates how, even in the presence of solid management of the CI, consistent use and parental support, language development is still delayed – seeming to indicate an explanation that goes beyond hearing loss.

However, I am not clear as to the argument being made for Child B (i.e., that there is no DLD). This child has weaker scores in all areas compared to Child C, and while it is true that there has not been optimal management of the CI etc., this does not necessarily preclude the presence of a DLD (p.10). While Child B is acquiring some signed language, there is no indication that this has developed at an age-appropriate level. Might it not be the case that even after the extrinsic factors are addressed, there might still be a delay?

It seems the relevant take-away is that a diagnosis of DLD would not be appropriate until certain indicators (i.e., implant use) had first been addressed (p. 13). Would this allow for differentiating between a DLD and a language disorder associated with deafness? It would be helpful to provide further clarification here.

Implications for Practice

In two paragraphs in the latter part of the Discussion (pp. 13-14), the authors consider clinical implications of this research. While I would agree that deafness and DLD can co-exist, how would a diagnosis of DLD as opposed to a language disorder associated with deafness make a difference in practice? What would differ with respect to subsequent support and follow-up from a speech language pathologist, teacher of the deaf, auditory-verbal therapist or other professional?

The starting place for any professional working with a child with a CI is ensuring management and consistent use of the technology. This is what allows for meaningful access to language, affording the opportunity to engage in contingently responsive communicative interactions with caregivers and others. Assuming access is in place, ongoing assessments alert practitioners as to whether development is proceeding appropriately. If not, action must be taken that typically includes a more explicit and targeted focus on language and communication. Does a DLD diagnosis change this?

In my experience, considering a child’s background (i.e., the extrinsic and intrinsic factors described in this paper) is standard practice in the field when making educational placement decisions, evaluating progress and planning interventions and support. That said, I am not clear on how the analysis in this paper adds to the considerations that have always been in place or how a more specific diagnosis of DLD would alter practice for children with CIs.

Additional Comments

Scores on the SIR are included in the Tables but are not explained in the text. Would these ratings also be related to making a diagnosis, and if so, how?

I wonder about the use of “speech-sound discrimination” to characterize the abilities assessed via the CAP and the NAMES (p.7). Might listening skills or auditory comprehension be more appropriate?

Editorial Points

  1. Mayberry & Squires (2006), Ching et al. (2017), Datta et al. (2016), Taha, Stojanovik & Pagnamenta (2021) and Christensen (2019) are cited in the text but not included in the References.
  2. Nicholas & Geers (2007), Spencer (2000), Van Christenen (2019), Wetherby et al. (2004) and National Institute on Deafness and Other Communication Disorders (2021) are listed in the References but not cited in the text.
  3. Jung & Houston (2020) on p.1 should be Jung et al.
  4. In text citations are not consistent re: use of et al. or listing all authors for publications with three or more authors (e.g., Bishop et al., 2017 on p. 2).

Author Response

Rev 2

To some extent, spoken language delays can be assumed for deaf children who receive cochlear implants, but research increasingly indicates that most do catch up with their hearing peers. That said, it is also an important and ongoing concern as to why some do not. The authors question whether these delays, at least in some cases, can be attributed to a developmental language disorder (DLD) that exists along with the hearing loss. They suggest that a diagnosis in this case can be made by considering four protective and risk factors that are both extrinsic (age at implantation and consistent device use, and quantity and quality of the parent-child interaction) and intrinsic (speech-sound discrimination, and non-verbal cognition) (pp. 4-5). As I understand their argument, if the language delay cannot be accounted for by an analysis of one of these factors, then it would be appropriate to posit DLD as an explanation, rather than a language disorder associated with deafness (p.3).  

While this is an interesting proposition, I have questions about the retrospective analysis, and the extent to which a diagnosis of DLD in children with CIs has utility for clinical and pedagogical practice.

Retrospective Analysis

I would like to see the authors say more about how these three case histories were chosen for analysis. What was the total participant pool from which these cases were drawn? What was the process by which these three were identified? Was there any relationship of the authors to this CI programme? As well, it would be important to explain why there is missing data for some time points in some areas (e.g., non-verbal assessments for Child A, Language for Child B in Year 5).

> We chose three children as stereotypical examples of the types of clinical and language profiles that are common in the CI setting. We have added some more information explaining this and missing data in the text. Bolded pages 5-6 

The description of Child A is clear and provides an example of how a child with a CI catches up to hearing peers. The depiction of Child C illustrates how, even in the presence of solid management of the CI, consistent use and parental support, language development is still delayed – seeming to indicate an explanation that goes beyond hearing loss.

However, I am not clear as to the argument being made for Child B (i.e., that there is no DLD). This child has weaker scores in all areas compared to Child C, and while it is true that there has not been optimal management of the CI etc., this does not necessarily preclude the presence of a DLD (p.10). While Child B is acquiring some signed language, there is no indication that this has developed at an age-appropriate level. Might it not be the case that even after the extrinsic factors are addressed, there might still be a delay?

>Yes it is possible, but taking all issues into the round there are too many doubts concerning the factors that are missing for ensuring language development. This clouds a diagnosis. This is not the case with Child C. We have added some more ideas to this section using the reviewer´s hesitancy but our main conclusion stands. Bolded sections p14.  

It seems the relevant take-away is that a diagnosis of DLD would not be appropriate until certain indicators (i.e., implant use) had first been addressed (p. 13). Would this allow for differentiating between a DLD and a language disorder associated with deafness? It would be helpful to provide further clarification here.

>Yes this is our main take-away. We have emphasised this more clearly. At bottom of page 13

Implications for Practice

In two paragraphs in the latter part of the Discussion (pp. 13-14), the authors consider clinical implications of this research. While I would agree that deafness and DLD can co-exist, how would a diagnosis of DLD as opposed to a language disorder associated with deafness make a difference in practice? What would differ with respect to subsequent support and follow-up from a speech language pathologist, teacher of the deaf, auditory-verbal therapist or other professional?

The starting place for any professional working with a child with a CI is ensuring management and consistent use of the technology. This is what allows for meaningful access to language, affording the opportunity to engage in contingently responsive communicative interactions with caregivers and others. Assuming access is in place, ongoing assessments alert practitioners as to whether development is proceeding appropriately. If not, action must be taken that typically includes a more explicit and targeted focus on language and communication. Does a DLD diagnosis change this?

In my experience, considering a child’s background (i.e., the extrinsic and intrinsic factors described in this paper) is standard practice in the field when making educational placement decisions, evaluating progress and planning interventions and support. That said, I am not clear on how the analysis in this paper adds to the considerations that have always been in place or how a more specific diagnosis of DLD would alter practice for children with CIs.

>the reviewer makes some very valid points and we have expanded the clinical and practical sections of the discussion accordingly. Two paragraphs page 15. We were aware of word count but agree these implications are important.

Additional Comments

Scores on the SIR are included in the Tables but are not explained in the text. Would these ratings also be related to making a diagnosis, and if so, how?

>thank you for mentioning this. We have provided more information on the SIR and include this now in discussion of each child.

I wonder about the use of “speech-sound discrimination” to characterize the abilities assessed via the CAP and the NAMES (p.7). Might listening skills or auditory comprehension be more appropriate?

>We are aware that this is an area with several terminological overlaps. We have provided some more information in the description of speech sound discrimination (SSD). We decided on SSD because the wider DLD research uses this term more that listening skills.

Editorial Points

  1. Mayberry & Squires (2006), Ching et al. (2017), Datta et al. (2016), Taha, Stojanovik & Pagnamenta (2021) and Christensen (2019) are cited in the text but not included in the References.

>Fixed

  1. Nicholas & Geers (2007), Spencer (2000), Van Christenen (2019), Wetherby et al. (2004) and National Institute on Deafness and Other Communication Disorders (2021) are listed in the References but not cited in the text.

>Fixed

  1. Jung & Houston (2020) on p.1 should be Jung et al.
  2. In text citations are not consistent re: use of et al. or listing all authors for publications with three or more authors (e.g., Bishop et al., 2017 on p. 2).

>Fixed